# Temporal and Spatial Changes of Soil Organic Carbon Stocks in the Forest Area of Northeastern China

**Shuai Wang** [1,2,3,4] 🔵**, Qianlai Zhuang** [3] 🔵**, Zijiao Yang** [1]**, Na Yu** [1,2,]*** and Xinxin Jin** [1,2,]***

[1] College of Land and Environment, Shenyang Agricultural University, Shenyang 110866, China; shuaiwangsy@163.com (S.W.); yangzijiao90226@163.com (Z.Y.)

[2] Key Laboratory of Northeast Cultivated Land Conservation, Ministry of Agriculture of the People's Republic of China, Shenyang Agricultural University, Shenyang 110866, China

[3] Department of Earth, Atmospheric, and Planetary Sciences, Purdue University, West Lafayette, IN 47907, USA; qzhuang@purdue.edu

[4] Key Laboratory of Ecosystem Network Observation and Modeling, Institute of Geographic Sciences and Natural Resources Research, Chinese Academy of Sciences, Beijing 100101, China

* Correspondence: sausoilyn@163.com (N.Y.); jinxinxin0218@syau.edu.cn (X.J.); Tel.: +86-024-8848-7155 (N.Y. & X.J.)

**Abstract:** Forest soil organic carbon (SOC) accounts for a large portion of global soil carbon stocks. Accurately mapping forest SOC stocks is a necessity for quantifying forest carbon cycling and forest soil sustainable management. In this study, we used a boosted regression trees (BRT) model to predict the spatial distribution of SOC stocks during two time periods (1990 and 2015) and calculated their spatiotemporal changes during 25 years in Liaoning Province, China. A total of 367 (1990) and 539 (2015) sampling sites and 9 environmental variables (climate, topography, remote sensing) were used in the BRT model. The ten-fold cross-validation technique was used to evaluate the prediction performance and uncertainty of the BRT model in two periods. It was found that the BRT model could account for 65% and 59% of SOC stocks, respectively for the two periods. MAP and NDVI were the main environmental variables controlling the spatial variability of SOC stocks. Over the 25-year period, the average SOC stocks increased from 5.66 to 6.61 kg m$^{-2}$. In the whole study area, the SOC stocks were the highest in the northeast, followed by the southwest, and the lowest in the middle of the spatial distribution pattern in the two periods. Our accurate mapping of SOC stocks, their spatial distribution characteristics, influencing factors, and main controlling factors in forest areas will assist soil management and help assess environmental changes in the region.

**Keywords:** forest soil; soil organic carbon stocks; spatial variation; environment variable

## 1. Introduction

Forest soil is an important component of forest ecosystems and one of the largest organic carbon pools in terrestrial ecosystems. It plays an irreplaceable role in regulating global carbon balance and slowing down the rise of atmospheric $CO_2$ concentration [1,2]. Dixon et al. [3] estimated the global forest soil carbon pool is 787 PgC based on literature review, accounting for 73% of the total global soil carbon. Soil structure is complex, its spatial distribution is uneven, and its spatial variability is large [4]. At present, the estimation of forest soil organic carbon (SOC) stocks has a large uncertainty, which contributes to an estimation error of annual carbon emissions from forest soils to the atmosphere (10 PgC) larger than the total industrial emissions (5.3 PgC) [5]. Therefore, a detailed investigation of forest SOC stocks shall help better quantify the global carbon balance.

The interaction of natural and human factors is the main cause of spatial variability of soil properties [6,7], including climate, topography, parent material, time, biology, and land use. Because

soil properties are affected by many environmental factors, it is difficult to predict soil properties accurately and efficiently at the regional scale [8]. As an efficient and low-cost method, digital soil mapping (DSM) technology is able to accurately describe the spatial variability of soil attributes in a region by using a limited amount of sampling data and environmental variables [9]. In order to accurately predict the SOC stocks in a region and analyze its key environmental factors, scholars have carried out a lot of researches [10–12]. In fact, the direct relationship between soil properties and environmental factors is non-linear and complex [13,14]. Therefore, machine learning algorithms which can effectively avoid this kind of problem are widely used in DSM mapping [11,15]. For example, on Barro Colorado Island, Panama, Grimm et al. [15] used a random forest model to simulate SOC stocks in 0–10 cm, 10–20 cm, 20–30 cm, and 30–50 cm layers, respectively. Giasson et al. [16] applied a multiple logistic regression model to predict the occurrence of soil types in southern Brazil. Dorji et al. [17] selected regression kriging and equal-area spline profile function to predict the SOC stocks at depths 0–5, 5–15, 15–30, 30–60, and 60–100 cm in montane ecosystems, Eastern Himalayas. In Denmark, Adhikari et al. [18] used regression kriging and 12 environmental variables to simulate the spatial distribution of SOC stocks at five soil layers. Were et al. [19] compared the performance of support vector regression, artificial neural network, and random forest models in predicting and mapping SOC stocks in the Eastern Mau Forest Reserve, Kenya.

Different from the DSM technology mentioned above, the boosted regression trees (BRT) model is a relatively new model based on tree development [20], which is rarely used in the study of temporal and spatial changes of SOC stocks, especially in forest areas. Compared with the single tree model, the BRT model has higher performance and efficiency [20]. In many previous studies, the BRT model is a reliable prediction model for soil properties [11]. The BRT model is the final prediction model generated by many simple tree models [21]. In this process, the model improves its performance through iteration and enhancement technology to reduce its fitting errors [19]. The BRT model can deal with imperfect data, such as outliers, missing values, and data interaction [21,22]. Thus, the BRT model is widely used in many fields, such as remote sensing [23], ecology [24], environmental science, epidemiology [25], fisheries science [26], and soil science [19,22]. However, the BRT model is rarely selected for the study of temporal and spatial changes of forest organic carbon stocks at regional scales.

In this study, the BRT model was used to estimate the temporal and spatial changes of SOC stocks in forest topsoil in Liaoning Province, China. The specific research objectives are to: (1) map the spatial distribution of soil organic carbon stocks from 1990 to 2015; (2) quantify the role of different environmental factors in the distribution of forest surface soil organic carbon; and (3) identify the spatiotemporal change pattern of SOC stocks during 25 years (1990–2015).

## 2. Materials and Methods

### 2.1. Study Area

Liaoning Province is located between 118°53′~125°46′E and 38°43′~43°26′N (Figure 1). Its land and sea areas are 145,900 km$^2$ and 152,200 km$^2$, respectively. Its elevation is higher in the East and west of Liaoning Province, but lower in the middle region. Mountainous and hilly areas are divided into east and west sides. The average elevations of mountainous areas in Eastern and Western Liaoning are 800 m and 500 m, respectively, and the elevation of the central Liaoning Province is 200 m. It has temperate continental monsoon climate, with hot and rainy seasons, and four distinct seasons. Its annual average temperature ranges from 7 to 11 °C, with the highest temperature above 30 °C above zero and the lowest temperature below 30 °C below zero. Its annual precipitation is 400–1100 mm. According to the second national land use classification system issued by the State Council of China in 2007, the forest area in the study area is divided into woodland, shrub woodland, sparse woodland, and other woodland. By the end of 2016, there were 63,440 km$^2$ of forestry land in Liaoning Province, including 46,410 km$^2$ of forest land (including 14,150 km$^2$ of economic forest), accounting for 73.1% of forestry land; 569 km$^2$ of sparse forest land, accounting for 0.9%; 2,275 km$^2$ of shrub forest area,

accounting for 3.6%; 14 190 km² of other forest land, accounting for 22.4%; and 31.8% of forest cover in the study area. Corresponding to the classification system of soil system in China [27], the main soil types of forest land in the study area are Primosols (36.7%) and Cambisols (33.4%), followed by Argosols (25.6%), Spodosols (4.1%), and Isohumosols (0.2%).

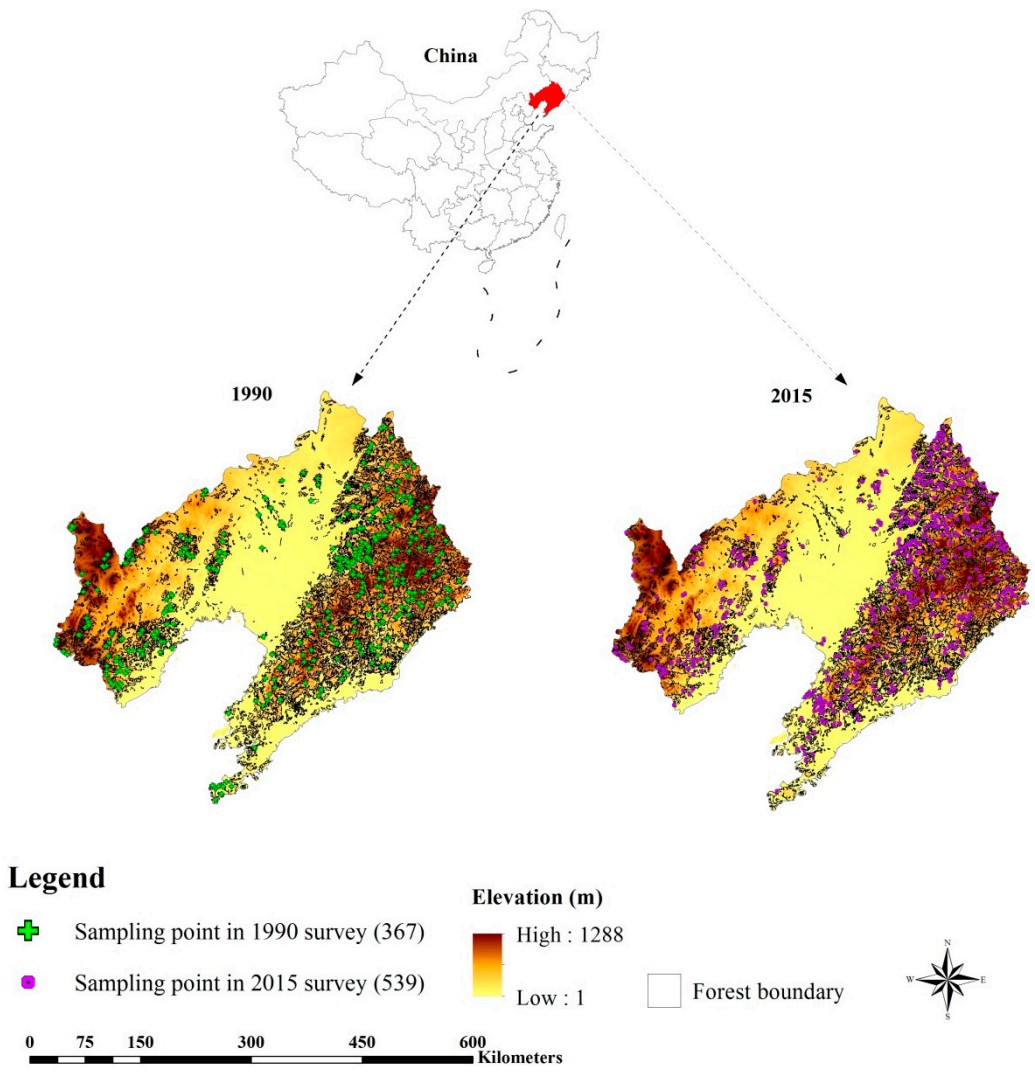

**Figure 1.** Location of sampling sites in Liaoning Province.

*2.2. Data Sources*

2.2.1. Soil Survey Data for 1990

The data of forest land soil properties in 1990 were obtained from the second National Soil Survey database of Liaoning Province [22]. A total of 367 forest soil data covering different soil types and parent material types were obtained. The database contains environmental information such as climate, topography, and parent material at sampling sites. In addition, this study focused only on the spatial variability of forest soil organic carbon (SOC) stocks in topsoil (0–30 cm), because we only extracted topsoil soil property data. For some sampling sites with missing soil bulk density (y), we used a Peod-transfer function (PTFs) and SOC content to estimate the BD. Specific formulas are as follows:

$$y = 1.52 - 0.08 * \sqrt{SOC} \left( R^2 = 0.74, \, p < 0.001 \right) \tag{1}$$

### 2.2.2. Soil Sampling in 2015

A total of 539 topsoil (0–30 cm) samples were collected, covering the forest topsoil of the whole province in 2015. The forest area is about 63,440 km$^2$, accounting for 59.5% of the total land area of the Liaoning Province. Therefore, it was difficult to collect soil samples in situ or in large quantities on the basis of sampling in 1990, especially in the rugged terrain with dense forest roads in the northeastern part of the study area. In order to accurately predict the surface SOC stocks in those regions, we applied a purposeful sampling method [28] to design a sampling strategy. Firstly, we selected the main environmental factors affecting the spatial variability of forest SOC stocks (mean annual temperature (MAT), mean annual precipitation (MAP), elevation, slope gradient, topographic wetness index (TWI), normalized difference vegetation index (NDVI)) and clustered the forest areas in the study area by using the fuzzy C-means clustering method [29]. Finally, 51 clustering landscape units were generated, and 10–12 sampling sites were collected in each unit. Longitudinal and latitudinal information was recorded by a hand-held GPS in each sampling point. The dead branches and leaves were removed from each sample point and 1 kg of mixed soil samples was collected. In the Center for Analysis and Testing, Shenyang Agricultural University, Shenyang, China, the samples were grinded and screened and the SOC content was finally determined using a C/N analyzer (Vario Max, Elementar Amerivas Ins., Germany). In addition, at each sampling site, soil samples were sampled with 100 cm$^3$ of undisturbed soil cores in the central part of the 0–30 cm soil layer and placed in an oven at 105 °C for 48 h to determine bulk density.

### 2.2.3. Environmental Variables

In this study, nine variables (elevation, slope gradient, TWI, MAP, MAT, B3, B4, B5, and NDVI) in three categories (Table 1) were selected as environmental variables to predict SOC stocks in those regions. Because these variables are collected from different departments with different precision, we use ArcGIS 10.2 software to resample them to grid data with 90 m spatial resolution. In addition, in order to facilitate modeling and analysis, we transformed the collected soil data and environmental variables into a unified projection coordinate system (Gauss Kruger).

**Table 1.** Environmental variables involved in the study.

| Class | Variable | Unit | Description | Resolution |
|---|---|---|---|---|
| Topography | Elevation | m | Absolute vertical distance to geoid | 90 m |
| | Slope gradient | ° | The Angle between the Normal Line of Ground Point and the Plumb Line | 90 m |
| | TWI | | Topographic wetness index | 90 m |
| Climate | MAT | °C | Mean annual temperature in 1990 and 2015 | 1000 m |
| | MAP | mm | Mean annual precipitation in 1990 and 2015 | 1000 m |
| Vegetation | NDVI | | Normalized Difference Vegetation Index | 30 m |
| | B3 | | Landsat5 Band 3 | 30 m |
| | B4 | | Landsat5 Band 4 | 30 m |
| | B5 | | Landsat5 Band 5 | 30 m |

Terrain-related variables are traditionally one of the widely used environmental variables in DSM. In this research, the digital elevation model (DEM) was downloaded from Geospatial Data Cloud site, Computer Network Information Center, Chinese Academy of Sciences [30]. ArcGIS 10.2 software was used to produce elevation and slope gradient, and then SAGA GIS software was used to obtain TWI. TWI generated by SAGA GIS can better reflect the details of the region and is more suitable for the analysis of this type of forest in the region.

MAT and MAP are the main climatic variables, obtained from the Meteorological Data Service Center of China [30]. Based on the observation data, historical data, and other auxiliary data, the spatial

distribution map of MAT and MAP needs to be made by using the climate data spatial interpolation Anusplin software [31] to predict the spatial distribution of MAT and MAP.

Remote sensing data covering from July to September in 1990 and 2015 (United States Geological Survey, USGS, 2017) were selected for modeling. Firstly, we used a polynomial geometric accuracy correction method to correct the image, and then applied an ENVI software to splice and cut the image. Finally, we selected Landsat TM band 3 (B3), Landsat TM band 4 (B4), and Landsat TM band 5 (B5) representing vegetation growth, cover, and biomass for modeling. In addition, the NDVI was calculated by using B3 and B4. The specific formulas are as follows:

$$NDVI = (B4 - B3)/(B4 + B3) \qquad (2)$$

### 2.3. Descriptive Statistics

Figure 2 summarized the descriptive statistics of SOC stocks and different environmental variables at sampling sits in 1990 and 2015. In 1990, SOC stocks ranged from 0.53 kg m$^{-2}$ to 16.13 kg m$^{-2}$, with an average of 9.1 kg m$^{-2}$. The average SOC stocks in 2015 were 9.44 kg m$^{-2}$. In 1990 and 2015, the skewness coefficients of SOC stocks were 0.89 and 1.21, respectively. The dataset presented a generalized skewness distribution. Logarithmic transformation of SOC stocks was carried out in two periods to make it conform to normal distribution.

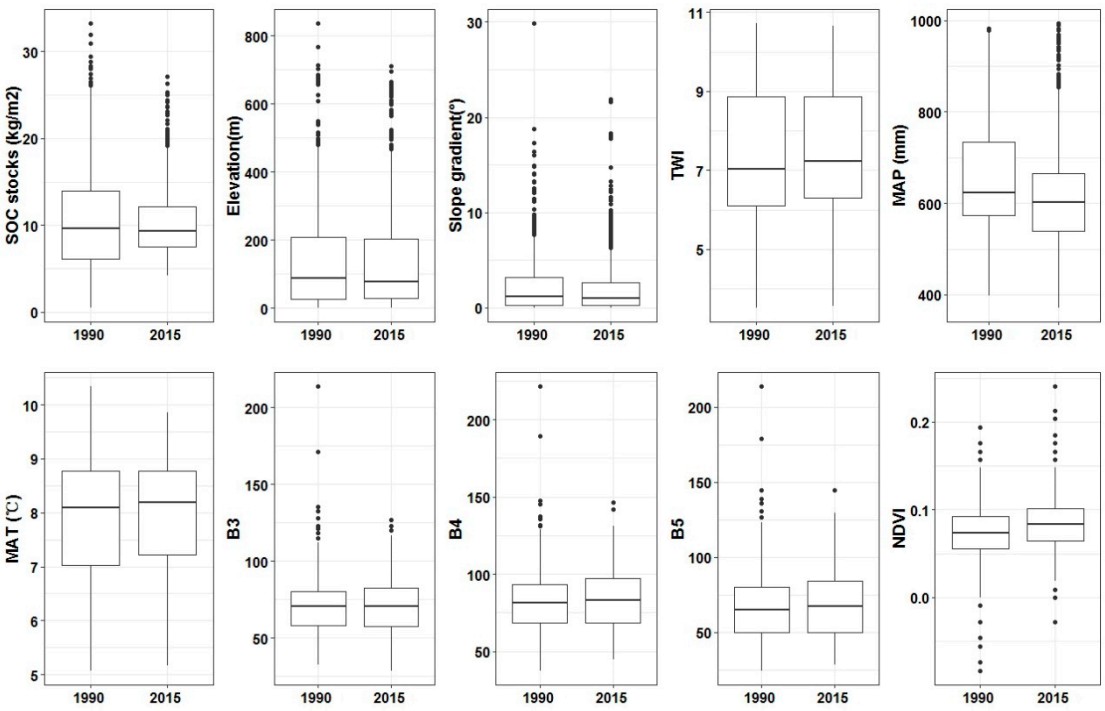

**Figure 2.** Boxplot of soil organic carbon (SOC) stocks and different environmental variable at sampling sits in 1990 and 2015.

Table 2 listed the correlation coefficients between SOC stocks and selected environmental variables over the periods in 1990 and 2015. Elevation, slope gradient, TWI, and MAP were positively correlated with SOC stocks in both periods. Correspondingly, SOC stocks were negatively correlated with MAT, B4, B5, and NDVI in both periods. In addition, there might be multiple collinearities between terrain variables and remote sensing image variables (Table 2). If this study used traditional statistical methods to predict SOC stocks, it might not be accurate. Therefore, the classical machine-learning algorithm-boosted regression trees (BRT) model was introduced to predict the spatial distribution of SOC stocks in this study.

**Table 2.** Relationships between observed SOC stocks (kg.m$^{-2}$) with environmental variables in 1990 and 2015 periods.

| Period | Property | SOC | Elevation | Slope Gradient | TWI | MAP | MAT | B3 | B4 | B5 |
|---|---|---|---|---|---|---|---|---|---|---|
| 1990 | Elevation | 0.31** | | | | | | | | |
| | Slope gradient | 0.18** | 0.49** | | | | | | | |
| | TWI | 0.23** | −0.56** | −0.69** | | | | | | |
| | MAP | 0.59** | 0.06 | 0.21** | −0.25** | | | | | |
| | MAT | −0.52** | −0.53** | −0.22** | 0.23** | −0.38** | | | | |
| | B3 | −0.17** | −0.16** | −0.18** | 0.07 | −0.17** | 0.19** | | | |
| | B4 | −0.28** | −0.19** | −0.21** | 0.06 | −0.27** | 0.22** | 0.71** | | |
| | B5 | −0.23** | −0.12** | −0.14** | −0.02 | −0.26** | 0.22** | 0.69** | 0.53** | |
| | NDVI | −0.46** | −0.13** | −0.08 | 0.03 | −0.35** | 0.41** | −0.37** | −0.07 | −0.04 |
| 2015 | Elevation | 0.37** | | | | | | | | |
| | Slope gradient | 0.14** | 0.46** | | | | | | | |
| | TWI | 0.18** | −0.57** | −0.71** | | | | | | |
| | MAP | 0.61** | −0.26** | 0.15** | −0.16** | | | | | |
| | MAT | −0.42** | −0.43** | −0.14** | 0.16** | −0.23** | | | | |
| | B3 | −0.05 | −0.07 | −0.12** | 0.12** | −0.14** | 0.09* | | | |
| | B4 | −0.19** | −0.04 | −0.16** | 0.07 | −0.27** | 0.19** | 0.64** | | |
| | B5 | −0.13** | −0.07 | −0.13** | 0.04 | −0.23** | 0.21** | 0.59** | 0.63** | |
| | NDVI | −0.39** | 0.06* | 0.03 | −0.13** | −0.34** | 0.35** | −0.31** | −0.19** | −0.18** |

Note: $p < 0.05$ shown in "*"; $p < 0.01$ shown in "**".

## 2.4. Boosted Regression Trees Model

In order to characterize the relationship between SOC stocks and all predictors, the BRT model developed by Friedman et al. [32] was used in this study. The BRT model is composed of two technologies: boosting and regression tree [19]. Boosting technology is an improvement based on the random gradient of decision tree [11]. It uses all samples at a time and changes the weight of samples at each round of training [20]. The goal of the next round of training is to find a function to fit the residuals of the previous round. It stops when the residual is small enough or reaches the maximum number of iterations set [23]. The BRT model can flexibly deal with soil environmental problems in complex landscape areas, and effectively avoid non-linearity and interaction [11]. Compared with the traditional regression model, the BRT model exhibits better prediction performance, especially in the spatial simulation of soil properties, which has been widely used in spatial prediction research [11,19,20]. In this study, we used the "dismo" package developed by Elith et al. [21] to construct the model in R language environment [33].

In the BRT model, four parameters (learning rate (LR), tree complexity (TC), bag fraction (BF), and tree number (NT)) need to be set by users [26]. LR represents the contribution of each tree in the model to the final fitting model [11]. TC is a direct predictor of tree depth and maximum interaction level [20]. BF represents the proportion of data used in each model [19]. NT is determined by LR and TC [23]. We test different combinations of these parameters by 10-fold cross-validation technology for obtaining the best prediction performance of the BRT model. Finally, we set LR, TC, BF, and NT to 0.0025, 9, 0.65, and 1200 in 2015, respectively. In 1990, LR, TC, BF, and NT were set to 0.0025, 9, 0.60, and 1000, respectively.

The BRT model was iterated 100 times, and the average standard deviation of 100 prediction results was calculated, which was used as an uncertain index to evaluate the prediction performance of SOC stocks.

In addition, we also calculated the relative importance (RI) of each environmental variable in simulating the spatial distribution of SOC stocks during the two periods. The RI of variables was measured based on the number of times a variable is selected for modeling, and weighted by the

square improvement of each segmentation and mean value of all trees. Finally, the average RI of 100 iterations of each variable in BRT model is converted to a percentage value.

## 2.5. Prediction Accuracy

In this research, we applied 10-fold cross-validation technology to test the prediction performance of the BRT model. Mean absolute prediction error (MAE), root mean square error (RMSE), coefficient of determination ($R^2$), and Lin's concordance correlation coefficient (LCCC) [34] were calculated using this technology in R software:

$$MAE = \frac{1}{n} \sum_{i=1}^{n} |p_i - o_i| \tag{3}$$

$$RMSE = \sqrt{\frac{1}{n} \sum_{i=1}^{n} (p_i - o_i)^2} \tag{4}$$

$$R^2 = \frac{\sum\limits_{i=1}^{n} (p_i - \bar{o}_i)^2}{\sum\limits_{i=1}^{n} (p_i - \bar{o}_i)^2} \tag{5}$$

$$LUCC = \frac{2r\partial_p\partial_o}{\partial_p^2 + \partial_o^2 + (\bar{p} + \bar{o})^2} \tag{6}$$

where $p_i$, $o_i$, $\bar{p}$, $\bar{o}$, $\partial_p$, and $\partial_o$ represent the predicted value, the measured value, the average value of the predicted, the average value of the measured, the variance of predicted value, and the variance of predicted and measured values at the sampling point $i$, respectively; $n$ denotes the number of sampling points; and r is Pearson correlation coefficient.

## 3. Results

### 3.1. Model Performance and Uncertainty

Ten-fold cross-validation technology was selected to verify the spatial prediction performance of the BRT model for SOC stocks during the two periods (1990 and 2015). The results showed that the BRT model presented higher $R^2$ and LUCC, and lower Mae and MRSE in both time periods (Table 3). Although there might be differences in sampling methods, experimental methods, and prediction models, our prediction accuracy was not inferior to that of previous studies. The average coefficient of variation (CV) of the 100 iteration maps predicted by the model for the two periods was shown in Figure 3. The average CVs in 1990 and 2015 were 7.3 and 13.2 kg m$^{-2}$, respectively.

**Table 3.** Prediction performance of the boosted regression trees (BRT) model iteration 100 times in the 1990 and 2015 periods.

| Period | Index | Min | Median | Mean | Max |
|--------|-------|-----|--------|------|-----|
| 1990 | MAE | 0.86 | 0.88 | 0.88 | 0.89 |
| | RMSE | 1.16 | 1.18 | 1.18 | 1.19 |
| | $R^2$ | 0.62 | 0.65 | 0.65 | 0.66 |
| | LCCC | 0.78 | 0.79 | 0.79 | 0.79 |
| 2015 | MAE | 0.65 | 0.66 | 0.65 | 0.67 |
| | RMSE | 0.87 | 0.88 | 0.88 | 0.89 |
| | $R^2$ | 0.57 | 0.59 | 0.59 | 0.61 |
| | LCCC | 0.72 | 0.73 | 0.73 | 0.74 |

Note: MAE, the mean error; RMSE, root mean squared error; LCCC, Lin's concordance correlation coefficient.

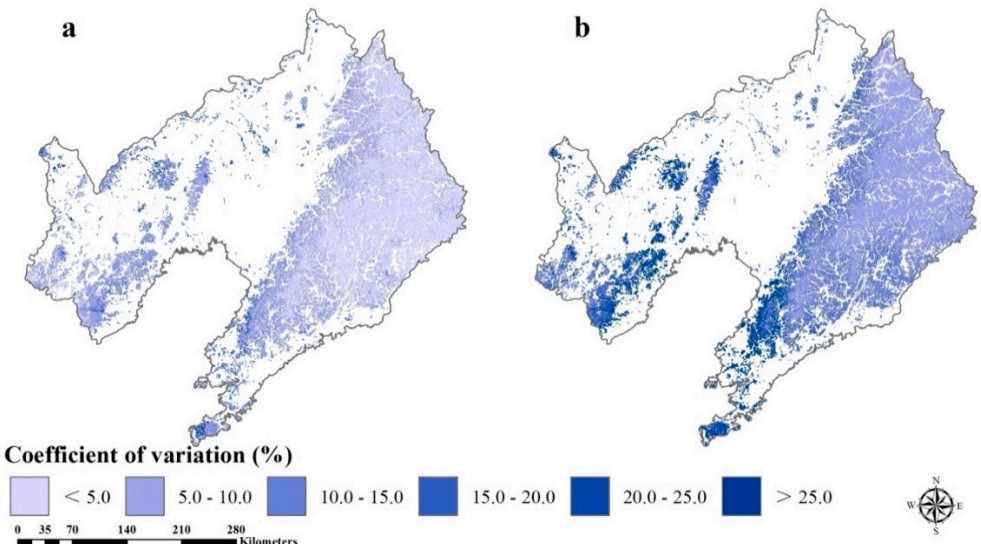

**Figure 3.** Coefficient of variation map of SOC stocks used boosted regression trees (BRT) in the 1990 (**a**) and 2015 periods (**b**).

### 3.2. Importance of Environmental Covariates

During the two research periods, the selected environmental variables showed different RI in predicting SOC stocks. MAP and NDVI were important environmental variables affecting SOC stocks in both periods. In 1990, the main factors affecting SOC stocks were remote-sensing-related variable, followed by climatic and topographic factors (Figure 4a). Correspondingly, the order of impacts was biological, topographic, and climatic factors in 2015 (Figure 4b).

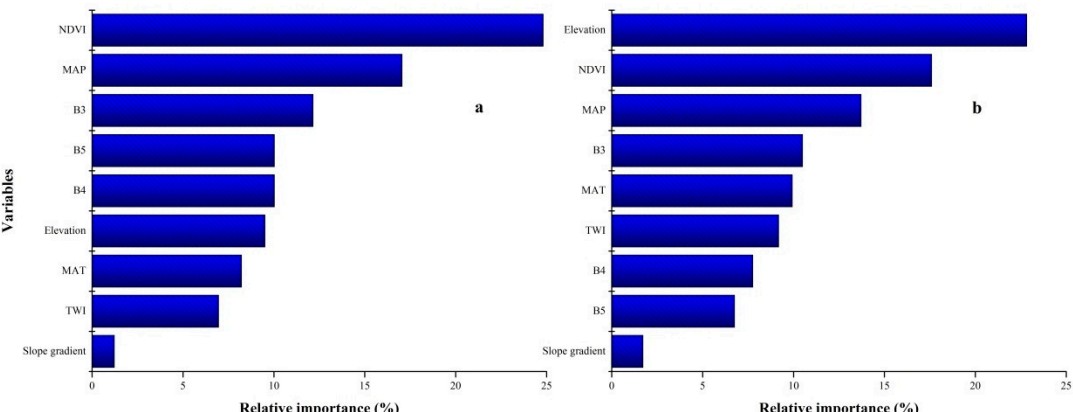

**Figure 4.** Relative importance of each variable as determined from 100 runs of the boosted regression trees (BRT) model in 1990 (**a**) and 2015 (**b**) periods.

### 3.3. Spatial Prediction of SOC Stocks

The BRT model was run 100 times to obtain the spatial distribution map of the average SOC stock in 1990 and 2015, respectively (Figure 5). From 1990 to 2015, the forest SOC stocks of Liaoning Province in Northeast China showed an upward trend (Figure 6a). The average SOC stocks increased from 5.66 kg m$^{-2}$ in 1990 to 6.61 kg m$^{-2}$ in 2015, an increase of 0.95 kg m$^{-2}$. A total of 83.7% of the study area increased SOC stocks, mainly in the eastern and western mountainous areas of Liaoning Province. In the past 25 years, the decline of forest SOC stocks was mainly concentrated in the central plain, accounting for 3.4% of the total area of the study area (Figure 6b).

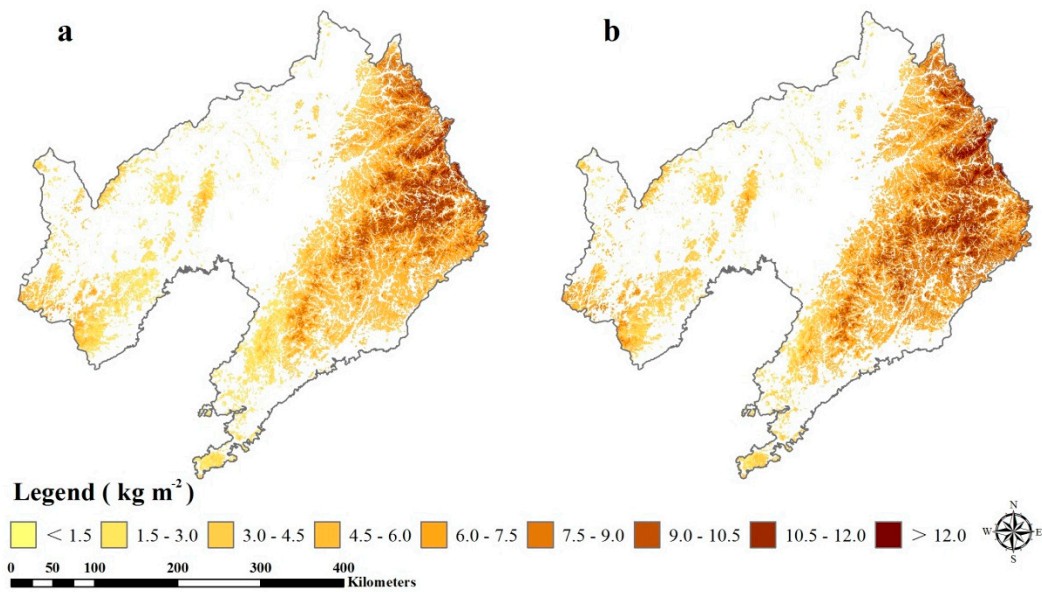

**Figure 5.** Spatial prediction of SOC stocks using the boosted regression trees (BRT) model in the 1990 (**a**) and 2015 (**b**) periods.

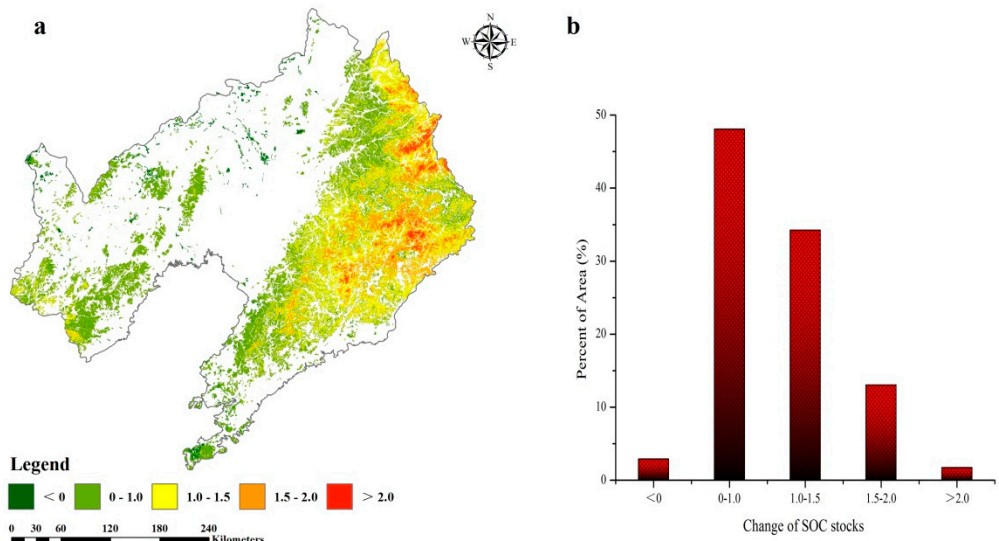

**Figure 6.** Spatial variation map of soil organic carbon (SOC) stocks (**a**) and percentages map of SOC stocks (**b**) for 25 years.

## 4. Discussion

### 4.1. Estimates of SOC Stocks

In 1990 and 2015, the forest SOC stocks showed an increasing trend in the northeast and southwest regions of the Liaoning Province, and a decreasing trend in the central region of the Liaoning Province. The areas with high SOC stocks were mostly concentrated in the eastern and western areas where forest resources were abundant. However, in the central plain areas where human activities interfered greatly, the SOC stocks tended to decrease. In 1990, we found that there was a high correlation between regional SOC stocks and vegetation variables, especially NDVI, which was confirmed by previous studies [22,35–37]. In Seoul Forest Park, Seoul, Republic of Korea, Bae et al. [37] found that the SOC stock in topsoil in 2013 was about three times that in 2003. By comparing the two periods of NDVI, it was found that the history of land use, the expansion of plant area, and the growth of plants are the main reasons for the increase of topsoil SOC stocks in the past 10 years. Except for vegetation-related

variables, SOC stocks also showed a close relationship with topographic variables, especially elevation variables, in 2015. Recent studies have pointed out that topographic variables, especially elevation variables, could be used as key environmental variables to effectively predict SOC stocks [11,22,36–40]. Tsui et al. [41] considered the effects of soil grade, vegetation type, and elevation on SOC stocks in subtropical wet volcanic ash in Yangmingshan National Park, Taipei City, Taiwan Province, China. In complex terrains, climate, vegetation type, and soil mineralogy change along elevation gradient [29]. Their results showed that elevation was a simple and effective prediction of SOC stocks. Different elevation gradients could form different hydrothermal conditions, which could affect the activity of microorganisms in soil and indirectly affect the accumulation and loss of organic matter, thus affecting the change of SOC stocks in the region.

We found that 3.4% of the whole study area showed a downward trend, mainly distributed in the plains of central Liaoning Province, and this part of the area was the main commodity grain producing area. The soil in this part of the area was strongly disturbed by human activities, and the land use pattern changed rapidly, which caused the decrease of organic carbon storage in the forest surface soil. Qi et al. [30] considered the topsoil SOC stocks in the same area. They believed that reclamation and other human activities were the main factors causing the decrease of SOC stocks in this area, which was confirmed in the studies of Bae and Ryu [37]. In most areas of the region, the forest SOC stocks showed an increasing trend, especially in the northeastern mountainous areas. The main reason is that this part of the region has been covered by forests and is seldom disturbed by man-made destruction, because the SOC stocks in this area showed an increasing trend, while the increase of SOC stocks in southwestern China can be attributed to the national policy of returning farmland to forests, leading to the increase of SOC stocks in this area. Wang et al. [22] used a BRT model and 9 environmental variables to simulate the spatial variations of SOC stocks in the forest-dominated area of Wafangdian, Liaoning Province, Northeast China. It was found that the increase of SOC stocks could be attributed to the implementation of the policy of returning farmland to forest and grass for many years in this area.

### 4.2. Controls of SOC Stocks

In both periods, vegetation-related variables were the most important influencing factors (Figure 4). This finding is consistent with the previous research conclusion. Bae and Ryu [37] considered that vegetation-related variables were closely related to the spatial distribution of SOC stocks. Vegetation is the main source of soil organic matter and controls the amount of organic matter entering the soil [36]. Among all the vegetation variables, NDVI and B3, representing the biomass and productivity of vegetation, respectively, were the most effective environmental variables to predict the main SOC stocks [42]. Our results showed that remote sensing and other related environmental variables play an important role in mapping regional forest topsoil SOC stocks. To some extent, B4 and B5 reflect the land use situation in those regions, while the SOC stocks under different land use patterns were significantly different. These results were consistent with the previous findings of Zhang et al. [43]. Bhunia et al. [44] used B3, B4, B5, and NDVI of Landsant 5 satellite to simulate the spatial variation of SOC stocks in India. Using remote sensing data has significantly improved the prediction of SOC stocks in this study.

The terrain determines the distribution of light, heat, and water, affecting the distribution of land vegetation types and the migration and transformation of SOC [18,22]. Different land-use and land-cover types and their changes also affect the input and accumulation capacity of SOC [17,37,38]. In addition, the spatial variability of soil properties and topographic variables all affect SOC changes [11]. In this study, we found that the RI of terrain-related variables in 2015 was higher than that in 1990 (Figure 4). Our results show that terrain variables were the main environmental factors and played an important role in the spatial variation of SOC stocks during these two periods. Elevation referred to the vertical distance of a place on the ground or a geographical object above or below sea level [20,45]. In mountainous vertical zones, with increasing elevation, the temperature gradually decreases, so the SOC density was vulnerable to this vertical zonality [45–47]. In addition, slope gradient could

significantly affect soil erosion, vegetation cover, and other factors, affecting SOC stocks. Liu et al. [45] discussed the effects of elevation and slope gradient on SOC stocks by fitting a linear equation. The results showed that elevation and slope gradient had significant effects. SOC stocks increased with the increase of elevation, but were negatively correlated with elevation at a certain height. The larger slope gradient tended to have higher SOC stocks. Tan et al. [46] studied the relationship between SOC stocks and natural factors in the topsoil of Ohio, Great Lakes, USA. It was found that the SOC stocks were closely related to the elevation in the forest cover area. Ramarson et al. [48] considered the influencing factors of SOC stocks in 1590 hectares of Madagascar Island. They concluded that elevation was the most influential factor when using a BRT model to simulate SOC stocks. In addition, the slope gradient and TWI were also the dominant topographic variables affecting the spatial variability of SOC stocks. The existence of slope gradient will affect the distribution of vegetation and soil layers. Especially in some low hilly and gentle slope areas, the increase of slope gradient led to thinning of soil layer and strong disturbance of human activities, which easily results in the loss of SOC [49]. However, when the slope gradient increased to a certain extent, which limited human activities and did not easily affect the accumulation of SOC, the impact of slope gradient on SOC was positively correlated [50]. TWI was a quantitative description of runoff length and confluence area. It was an effective index to measure soil water content. TWI and soil moisture change synergistically. Soil moisture was also very high in areas with large TWI. Huang et al. [50] and Raduła et al. [49] considered that the relationship between TWI and soil moisture was linear, and the spatial change of TWI was proportional to the spatial change of soil moisture. Therefore, it was feasible to use TWI to reflect the level of soil moisture. In previous studies, it was found that SOC also had a significant positive correlation with soil moisture [45,46,49,50]. They pointed out that the increase of SOC could promote aggregate formation, improve soil structure, and consequently increase soil water retention capacity, and reduce the loss of surface soil water due to evaporation or infiltration into the lower layer [49,50]. Therefore, TWI can be used as an effective factor to predict SOC stocks.

Climate variables were the key environmental variables affecting SOC stocks [11,20]. Previous studies [18,22,30,46,48] had revealed that climate-related variables were closely related to SOC stocks, especially on the regional scale [18], and MAP and MAT were considered to be the main climate variables controlling the spatial variability of SOC stocks. The correlation between MAP and SOC stocks decreased with the decrease of spatial scale, while the correlation between MAP and SOC stocks was not obvious with the change of spatial scale, and there was a strong regional difference [42]. SOC stocks were mainly affected by the combined effect of MAP and MAT [51]. In this study, the explanatory ability of climate variables to the variation of SOC stocks was about 20% during two periods, and the explanatory ability gradually decreased with the decrease of scale. With the increase of MAP, the annual output of organic matter increases correspondingly, so the organic carbon quality entering the soil also increases [52]. In addition, the RI of climate variables was offset to some extent by related environmental factors such as topography [42]. For instance, areas with different elevations have different hydrothermal conditions, affecting SOC stocks in mountainous areas.

### 4.3. Uncertainty in Current Research

The results showed that, although the BRT model had excellent prediction performance in predicting SOC stocks during the 1990 and 2015 periods, there were still some uncertainties in this study. First, while the soil data were obtained from the historical dataset of the Second National Soil Survey Database of Liaoning Province in 1990, the whole dataset come from different departments, which might have sampling errors or experimental errors. Second, because some bulk density data were missing in 1990, we used a Pedo-transfer function (PTFs) (Formula 1) to predict the missing bulk density. Nevertheless, the soil was not homogeneous in those regions and the distribution estimates of land cover types might have also been biased. Finally, this study was limited to the estimation of SOC stocks in the topsoil (0–30 cm), which would have underestimated the total SOC in this area. Especially in the forest covered areas, the SOC stocks are usually stored at deeper layers in the forest soils.

## 5. Conclusions

We used a BRT model to simulate the spatial distribution of forest topsoil (0–30 cm) SOC stocks in Liaoning Province, China, and determined their key environmental variables. We found that, in the past 25 years, the average value of SOC stocks has increased from 5.66 kg m$^{-2}$ to 6.61 kg m$^{-2}$ in this region. Surprisingly, the SOC stocks showed a decreasing trend in the central plain area of the study area, accounting for 3.4% of the total area. It was also found that NDVI and MAP were two key environmental variables that affect the spatial distribution of SOC stocks in the two periods. Overall, this study provided a more accurate prediction of regional SOC, which shall help ecological restoration, forest protection, and environmental management in the forest areas of the region.

**Author Contributions:** Experimental design by S.W. and N.Y.; sample collection and experiment by Z.Y., S.W. and X.J.; manuscript writing by S.W.; manuscript modification and editing by Q.Z. and S.W.

**Acknowledgments:** The study was supported by the National Key Technology R& D Program of China (Grant No. 2016YFD0300807); Young scientific and Technological Talents Project of Liaoning Province (Grant No. LSNQN201910 and LSNQN201914); and the National Basic Research Program of China (973 Program) (Grant No. 2011CB100502).

**Conflicts of Interest:** The authors declare no conflict of interest.

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
