# Peer review of "Temporal and Spatial Changes of Soil Organic Carbon Stocks in the Forest Area of Northeastern China"

_forests, doi:10.3390/f10111023_

Round 1
Reviewer 1 Report
Autors demonstate that BRT model had good prediction performance for the spatial and themporal distribution of SOC stocks. Minor remarks in the article.

Author Response
Reviewer 1
General comments:
Autors demonstate that BRT model had good prediction performance for the spatial and themporal distribution of SOC stocks. Minor remarks in the article.
Reply: We appreciate your help and your patience. With this submission, we provided a version (marked) of the revised manuscript. Responses to reviewers’ comments on the manuscript of marked are detailed below.
Specific Comments:
L168, 170 Positive or negative?
Reply: This is a positive correlation, and we clarified this in this revision. L186-188
L175 and pH???
Reply: We have corrected this. L193-194
L176 Table 1. How can it be? The steeper the slope, the more erosion the greater the loss of carbon?
Reply: Your comment is correct, but in the actual field sampling process, we found that it is difficult to stand on the steep slope and collect samples, so the soil samples we collected are concentrated on the gentle slope.
Figure 3 SD depends on the size of the average. This map may be interpreted like "in 1990, CO data were overestimated (for example, due to a systematic error in the method). The best way to use variation coefficient (or other non-dimension estimation).
Reply: Based on your comments, we changed the SD into a coefficient of variation and modified it in the manuscript. See the Figure 3

Reviewer 2 Report
This is a good article and is comprehensive in its coverage of "Spatial Changes of Soil Organic Carbon Stocks in Forest Area".
The title is accurate and the main research topic of the manuscript is of interest for many researchers and relevant to current issues in “Forests”, so, the paper is relevant to the aims and scope of the journal.
The paper is well structured and the conclusions are quite accorded with the results and discussion and it is consistent with other studies.
In order to improve the article, please:
- review the English of some sentences;
- figure 1 has to be completely redone:
- the colors do not respect the international relief conventions (brown tones),
- it is difficult to distinguish the two symbols (1990 survey e 2015 survey),
- the forest boundary is not visible;
- add some recent literature that is missing (little literature from non-Chinese authors)
Author Response
Reviewer 2
General comments:
This is a good article and is comprehensive in its coverage of "Spatial Changes of Soil Organic Carbon Stocks in Forest Area".
The title is accurate and the main research topic of the manuscript is of interest for many researchers and relevant to current issues in “Forests”, so, the paper is relevant to the aims and scope of the journal.
The paper is well structured and the conclusions are quite accorded with the results and discussion and it is consistent with other studies.
Reply: We appreciate your help and your patience. With this submission, we provided a version (marked) of the revised manuscript. Responses to reviewers’ comments on the manuscript of marked are detailed below.
Specific comments:
In order to improve the article, please:
Review the English of some sentences;
Reply: We found a professional language staff to retouch the language of the whole manuscript.
Figure 1 has to be completely redone: the colors do not respect the international relief conventions (brown tones), it is difficult to distinguish the two symbols (1990 survey e 2015 survey), the forest boundary is not visible;
Reply: Based on your comments, we have modified Figure 1. See Figure 1.
Add some recent literature that is missing (little literature from non-Chinese authors)
Reply: Based on your comments, we have added some recent literature. L530-539